# Contact evolution of dry and hydrated fingertips at initial touch

**Gokhan Serhat**[1]☯*, **Yasemin Vardar**[1,2]☯, **Katherine J. Kuchenbecker**[1]

**1** Haptic Intelligence Department, Max Planck Institute for Intelligent Systems, Stuttgart, Germany,
**2** Department of Cognitive Robotics, Delft University of Technology, Delft, CD, The Netherlands

☯ These authors contributed equally to this work.
* serhat@is.mpg.de

**Data Availability Statement:** The anonymized data will be publicly available in Max Planck Open Access repository upon acceptance.

**Funding:** The authors received no specific funding for this work.

## Abstract

Pressing the fingertips into surfaces causes skin deformations that enable humans to grip objects and sense their physical properties. This process involves intricate finger geometry, non-uniform tissue properties, and moisture, complicating the underlying contact mechanics. Here we explore the initial contact evolution of dry and hydrated fingers to isolate the roles of governing physical factors. Two participants gradually pressed an index finger on a glass surface under three moisture conditions: dry, water-hydrated, and glycerin-hydrated. Gross and real contact area were optically measured over time, revealing that glycerin hydration produced strikingly higher real contact area, while gross contact area was similar for all conditions. To elucidate the causes for this phenomenon, we investigated the combined effects of tissue elasticity, skin-surface friction, and fingerprint ridges on contact area using simulation. Our analyses show the dominant influence of elastic modulus over friction and an unusual contact phenomenon, which we call friction-induced hinging.

## Introduction

When one touches an object, the initial contact provides many clues about its surface properties. As finger-object contact develops, the deformation of the finger pad evokes tactile cues related to surface topography, friction, and other attributes even when there is no bulk lateral movement [1–3]. This deformation is influenced by the finger's geometry (i.e., size, shape, dimensions of fingerprint ridges, tissue layer thicknesses) as well as its material properties (i.e., elastic modulus, Poisson's ratio) [4, 5]. Characteristics of both the surface (e.g., roughness and hardness) [6] and the skin-surface interface (e.g., static and kinetic friction coefficients) [7, 8] also affect finger deformation during contact.

The complexity of touch mechanics largely stems from the interdependence between the elastic properties of the skin and the frictional forces applied to the finger by the contact surface. A change in the elastic moduli alters the total friction force due to the alteration of the real contact area [9], while a change in the friction coefficient results in a different elastic deformation due to the changed tangential forces that resist stretching [8]. In addition, these two variables are both influenced by moisture, an important external factor. An increase in the

**Competing interests:** The authors have declared that no competing interests exist.

fingertip moisture level softens the finger [10–12] while also changing the skin-surface friction coefficient [13–15]. The fingerprint ridges further complicate touch mechanics, as they cause non-uniform contact area as well as additional effects on the elastic behavior [4, 16] and friction [17, 18] when moisture is involved.

The evolution of the finger contact area has been well investigated using optical imaging techniques for static [4, 6] and dynamic [19] normal loading, as well as immediately before and during tangential slip [16, 20]. However, limited attention has been paid to finger contact evolution at initial touch under different hydration conditions, which have been reported to have a major influence on the skin deformation mechanics during tangential loading [10, 15, 16]. One recent study even showed that the fingerprint ridges systematically absorb and release moisture to regulate grip [18]. Moisture's dual impacts on the finger's elastic properties [10, 11] and frictional interactions [13–15] may both change the finger contact area. Moreover, hydration forms liquid bridges and films, causing capillary adhesion that also affects the resultant contact area [21]. Therefore, it is not straightforward to investigate the parametric influences of hydration on elastic properties and friction during contact area evolution via experimental measurements. Herein, we show that computational approaches can be utilized as alternatives.

The undulating geometry of the fingerprint ridges greatly increases the modeling complexity of finger-surface contact mechanics [22]. Furthermore, the significantly higher elastic modulus of the thin outermost skin layer (stratum corneum) increases the influence of its geometry on the results. These factors limit the usage of many analytical contact analysis methods that assume the finger has hemispherical or ellipsoidal geometry and/or uniform material properties. Alternatively, finite element analysis (FEA) has been prevalently utilized to analyze the parametric effects of physical parameters on finger contact. For instance, Maeno et al. [23] constructed a two-dimensional (2D) finite element model of a human fingertip to analyze its elastic properties. They found reasonable values for the Young's moduli of the epidermis, dermis, and subcutaneous tissue by comparing experimental measurements with the FEA model's predictions for how contact area would change with normal force. Later, Dandekar et al. [24] built three-dimensional finite element models of human and monkey fingertips. They then computed the elastic moduli that lead to correct deformations during static indentation, showing that layered finger models are necessary to replicate the deformations they experimentally observed. In another effort, Maeno et al. [25] proposed the use of FEA to estimate the friction coefficient between a finger-like elastic object and a planar surface. They showed that shear strain within the elastic body varies with the friction coefficient value. Leyva-Mendivil et al. [26] investigated the influence of the stratum corneum on the finger deformation mechanics. Despite its thinness, they showed that the large Young's modulus of the stratum corneum significantly influences the strain magnitude and strain direction within all finger layers. Even with the aforementioned efforts, the combined parametric influences of different elastic modulus and friction coefficient values on the finger deformation mechanics during initial touch has remained unexplored.

This study investigates the contact evolution of dry and hydrated fingertips at initial touch to understand the roles of different physical factors in this process. Using optical imaging, we measured the gross and real contact areas of two participants' finger pads while they were actively pressing on a glass surface up to 1 N. The experiments were conducted for three skin conditions: dry, hydrated with water, and hydrated with glycerin. These conditions were selected to match dry and moist finger interactions that humans experience daily. For example, humans frequently interact with water ($H_2O$) when washing their hands, showering, and doing dishes. Sweat, which is naturally excreted from the skin and has been shown to regulate finger grip [6, 18], also substantially contains water. Moreover, glycerin is a non-toxic chemical

compound ($C_3H_8O_3$) generally used in cosmetic creams, soaps, and other pharmaceutical skincare products to moisten dry skin.

As the measured contact area evolution reflects the combined effects of moisture, friction, and adhesion, we also performed finite element simulations to analyze how the finger's softness and the skin-surface friction affect the contact line (1D analogue of 2D contact area) in a controlled numerical environment. Although the evolution of gross contact area was similar for the three hydration conditions, the real contact area evolution showed drastic differences for finger pads hydrated with glycerin compared to other two conditions for both participants. The simulation results showed that the elastic moduli influence both gross and real contact area more significantly than friction. Because new fingerprint ridges make contact at particular pressing force values, the gross contact line vs. pressing force graphs exhibited sudden jumps that were also experimentally observed for the gross contact area. Around these force values, we also noticed that the gross contact line length for higher friction momentarily exceeds the one observed for lower friction. This occurrence is counter-intuitive since increasing friction has been reported to decrease the gross contact line [25]; we name the responsible physical phenomenon friction-induced hinging.

## Results

### Contact area measurements

Finger pad images of two participants (S1 and S2) were collected using the frustrated total internal reflection (FTIR) technique (see Materials and Methods for details). Two participants actively pressed their dominant index finger at the center of a glass surface at a fixed contact angle (approximately 0°); they gradually increased their pressing force up to 1 N in 5.76 seconds (see Fig 1B for a schematic of the apparatus). They performed experiments in three skin hydration conditions: dry, hydrated with water, and hydrated with glycerin. Each condition was repeated for four times. Fig 1 shows raw images collected from one participant in one trial for each condition as a function of the normal force, plus the corresponding binary images at 1 N.

As the pressing force rate significantly affects finger deformations [6, 27], the recorded pressing force data from each measurement were fitted to a first-order linear function ($F_n(t) = at + b$), where $a$ is the slope, $b$ is the intercept, and $t$ is time. The resulting fits ($R^2 > 0.92$) and the recorded force data are shown in Fig 2A, and the values of the best fit slopes appear in Fig 2B.

The collected finger pad images were processed by following the procedures explained by Lévesque and Hayward [28] (see Materials and Methods for details). The real contact area, $A_r$, of the finger pad in each image was calculated by summing the number of black pixels in the binarized image and multiplying that number by the pixel area. The gross contact area, $A_g$, was taken to be the area of the polygon fitting the outer contour of the binary fingerprint. Fig 2A shows the evolution of gross and real contact areas of both participants for all trials, along with their proportion, $A_r/A_g$. For direct comparison with the simulation results, the gross and real contact lines were also calculated along the line passing through the fingerprint centroid perpendicular to the longitudinal finger axis (see S1 Fig).

The data for gross and real contact areas for each trial were fitted to a power-law [4, 22] expression of the following form:

$$A = kF_n{}^m \tag{1}$$

where $A$ is contact area, $F_n$ is applied normal load, $k$ is load coefficient, and $m$ is load index. In this equation, $k = k_g$ and $m = m_g$ when $A = A_g$, and $k = k_r$ and $m = m_r$ when $A = A_r$. The

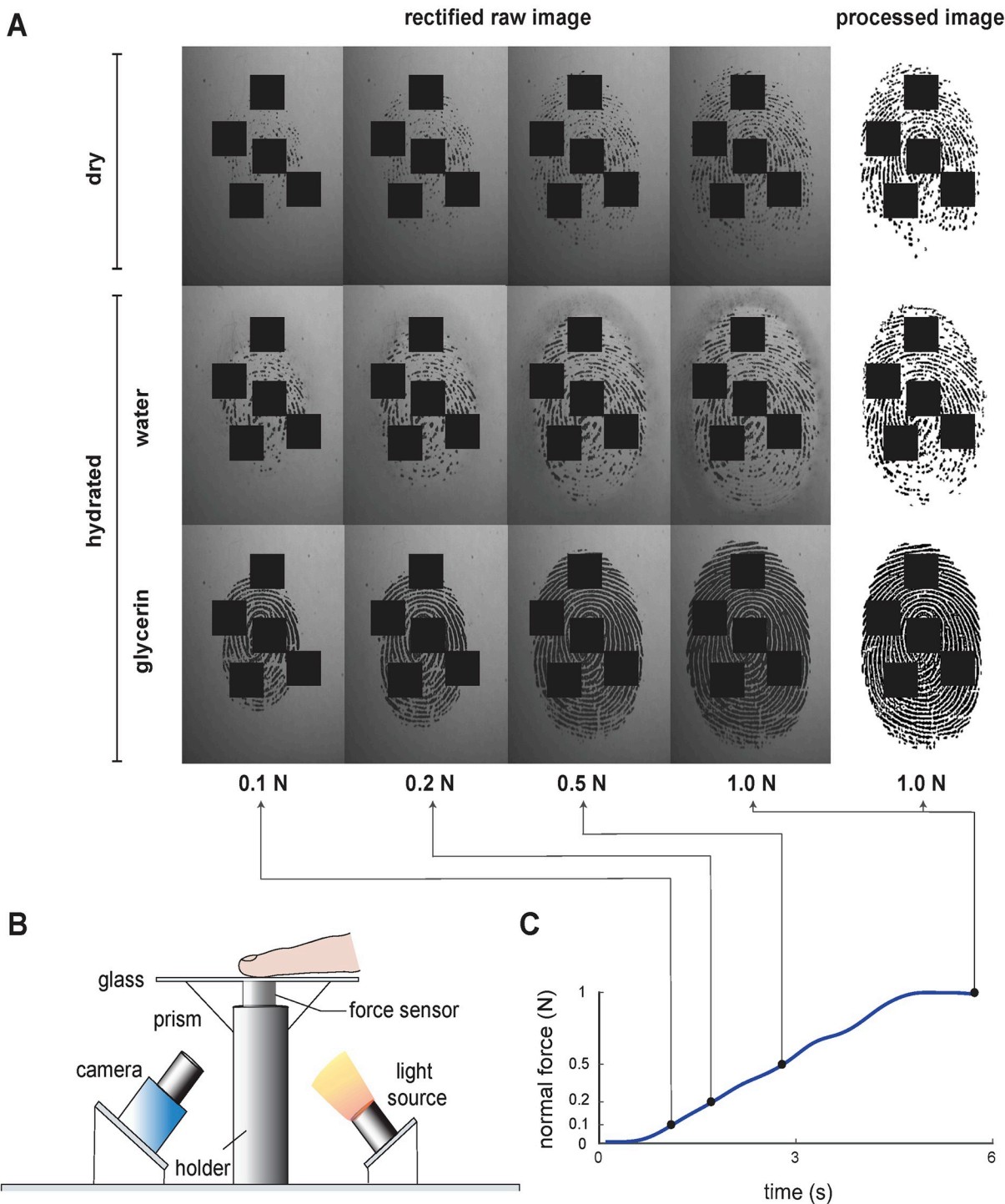

**Fig 1. Illustration of the experimental setup and collected finger pad images in each experimental condition.** (A) Example raw finger pad images collected in dry, water-hydrated, and glycerin-hydrated conditions as a function of normal force, and their corresponding processed image at 1 N. Four square regions of each fingerprint are obscured to protect the identity of the participant. (B) Schematic of the experimental apparatus. (C) Example finger normal force evolution during one trial. These sample finger pad images and force data were collected from S1.

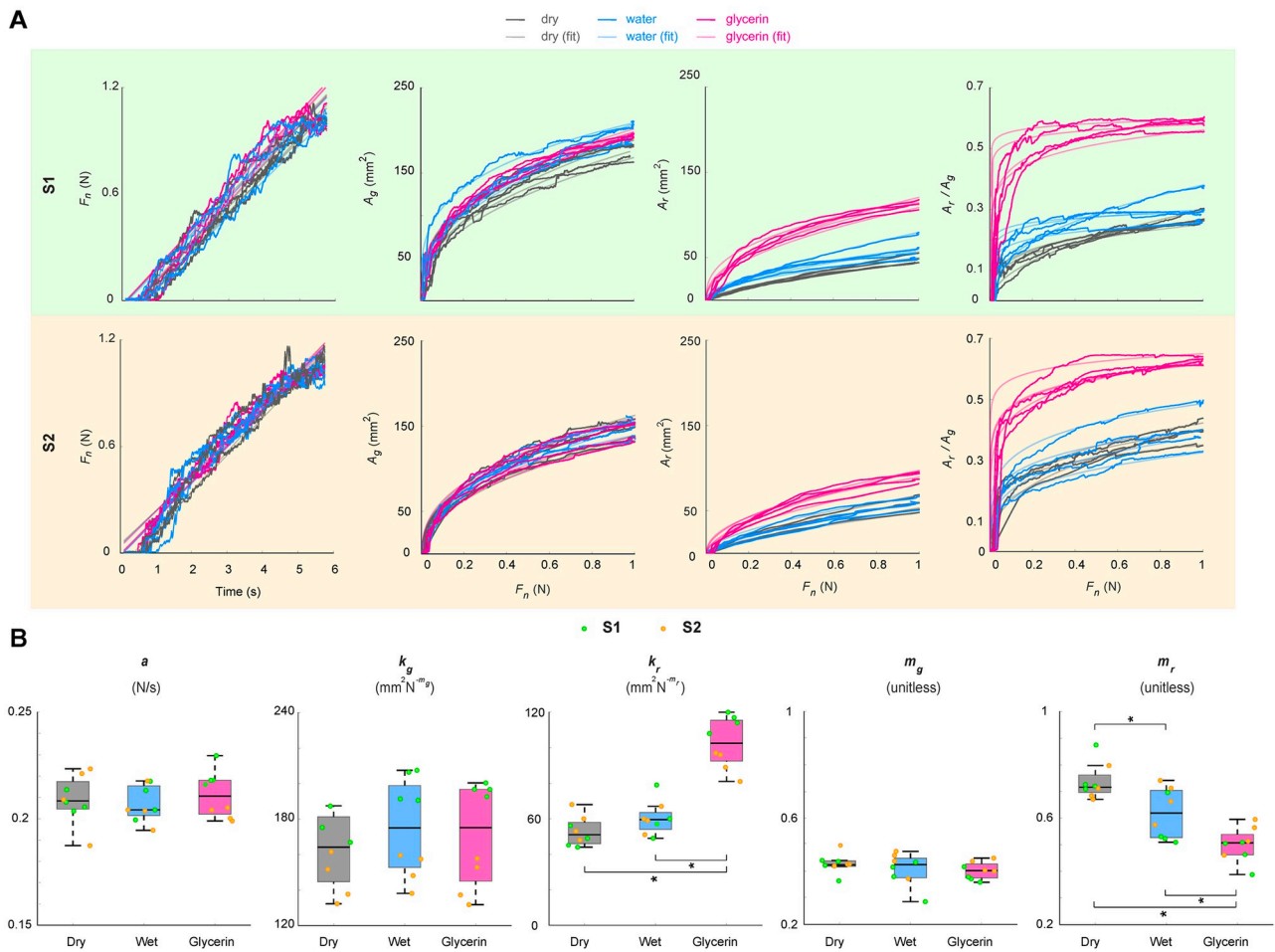

**Fig 2. Results of the contact area measurements.** (A) The normal forces applied by the two subjects (S1 and S2) for the dry, water-hydrated, and glycerin-hydrated conditions and the corresponding evolution of gross and real contact area, along with their proportion $A_r/A_g$, as a function of applied force. The traces are color coded by experimental condition. (B) Boxplots of the slope $a$ obtained by fitting the linear function $\dot{F}_n(t) = at + b$ to pressing force versus time and the parameters $k_g$, $k_r$, $m_g$, and $m_r$ obtained by fitting the power function $A = kF_n{}^m$ to gross and real contact area versus applied force. The center lines show the medians; box limits indicate the 25th and 75th percentiles. The whiskers extend to 1.5 times the interquartile range. The data points represented by the boxes are also plotted and color coded for each participant. The connected brackets with stars ($\star$) mark statistically significant pairwise differences.

resulting fits ($R^2 > 0.93$) appear with the data in Fig 2A, and the values of the best fit parameters are shown in Fig 2B. Increasing the load from zero increases the contact area proportional to $k$ because more skin comes into contact. Hence, a higher load coefficient $k$ indicates a larger increase in contact area as a function of the applied normal force. The load index exponent $m$ shows the shape of the contact evolution. For the real contact area of dry fingers, this exponent is close to one, causing a close-to-linear increase in the real contact area. However, when the finger is hydrated, the real contact area evolves in a more non-linear way. For small pressing forces, the gross and real contact areas grow rapidly due to the ellipsoidal geometry of the finger. The curves flatten out at higher loads.

General linear mixed models were created to test the fixed effect of hydration condition on $a$, $k_p$, $k_g$, $m_r$, and $m_g$. The subject number and measurement order were specified as random effects in the analysis. The loading force slopes did not differ across the three tested conditions.

The hydration condition significantly affected real contact area parameters ($k_r$: $F(2, 14) =$ 56.45, $p < 0.001$ and $m_r$: $F(2, 14) = 26.22$, $p < 0.001$) but not those for gross contact area ($k_g$ and $m_g$). A sequential Bonferroni-corrected post hoc test showed that the $k_r$ values for the glycerin condition were significantly higher than the other conditions ($p < 0.001$). On the other hand, the $m_r$ for the glycerin condition was significantly lower than the other conditions ($p < 0.01$), and $m_r$ for the dry condition was significantly higher than the water one ($p < 0.01$).

### Parametric finite element simulations

We utilize finite element analyses to investigate the influences of stratum corneum elastic modulus and skin-surface friction coefficient on the deformation mechanics of pressing touch. The simulations offer the advantage of showcasing the individual and combined effects of the desired parameters without being influenced by external factors. In contrast, experiments are often affected by uncontrollable variables and permit the measurement of only a limited subset of the quantities of interest. The investigation of collective effects is particularly important since physical variables (e.g., moisture) and human characteristics (e.g., age) can alter the skin's elastic properties and friction simultaneously on both short and long timescales.

**Form of contact forces.** We initially investigate the form of the contact forces that govern the finger deformation. Fig 3A shows the deformed fingertip for the nominal elastic moduli (see Table 2 in Materials and methods for the chosen material properties), a modified friction coefficient of $\mu = 1.0$ mm$^{-1}$, and a pressing force of $F_n = 1.0$ N. Fig 3B plots the normal and tangential force distributions along the contact line for this scenario. The modeling of multiple tissue layers allows the simulations to capture the finger's deformation mechanics different from a homogeneous body. The balloon-like deformation profile originates from the significantly larger elastic modulus of the stratum corneum compared to the interior soft tissue layers. The elliptical finger cross-section, the fingerprint ridges, and skin-surface friction yield characteristic normal and tangential force profiles. The nodal normal force profile has a wavy pattern, which reflects the presence of the fingerprint ridges. There is also a larger-scale trend that manifests the ellipse-like geometry of the finger exterior. The friction forces are not significant at the center because of the small tangential displacements there. The tangential force profile

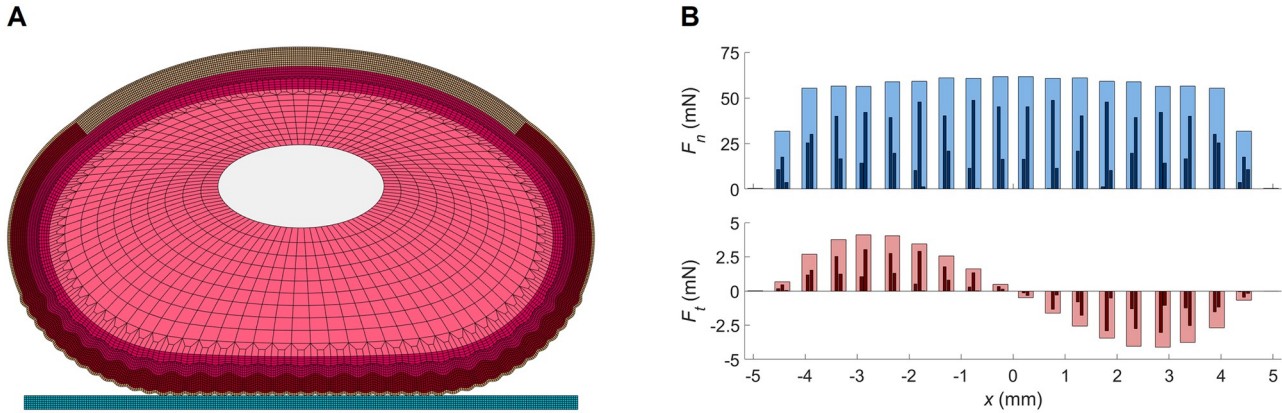

**Fig 3. An instance of the fingertip FE model in contact with a frictional flat surface and the corresponding contact force profiles.** (A) Deformed finger profile for the nominal elastic moduli, friction of $\mu = 1.0$ mm$^{-1}$, and 1.0 N pressing force. (B) Normal and tangential force ($F_n$ and $F_t$) distributions along the contact line for the central region of the fingertip, which includes 20 ridges with a total of 46 nodes in contact (two or three per ridge). The total forces applied on each ridge are shown with light colors behind the nodal forces. The contact forces on the two outermost ridges are almost negligible, and there are no contacts outside the depicted region. The sum of the absolute values of the nodal friction forces at this simulation step is 0.047 N.

has higher amplitudes in the middle regions between the center and endpoints of the contact. The effect of friction reduces toward the outer edges of the contact region due to decreasing normal forces and the fact that newly contacting ridges are in lateral equilibrium at contact.

**Contact lines for different stratum corneum moduli and friction coefficients.** The presence of friction influences fingertip deformation mechanics even in pure normal contact when there is no bulk lateral load or sliding motion. As the contact force increases, the Poisson effect causes the tissue to laterally deform. When friction is present, this deformation is lessened due to lateral resistive forces, while the tissue can spread out further in the absence of friction, thereby increasing contact line length. We explore this phenomenon by altering the modified friction coefficient ($\mu$) from 0.0 to 2.0 mm$^{-1}$ in increments of 0.5 mm$^{-1}$. Because the elastic modulus of exterior skin differs with moisture level and other environmental factors [29], we also investigate the influence of stratum corneum softening on the fingertip deformation mechanics. We conduct analyses for five $E_{sc}$ values: 1.0 (nominal), 0.2, 0.1, 0.067, and 0.05 MPa. The latter four values represent 5, 10, 15, and 20 times softer stratum corneum layers, respectively. For all combinations of the selected values, the gross and real contact lines (1D analogues to gross and real contact areas) were calculated while the simulated fingertip was pressed into the rigid surface.

Fig 4 presents (A) gross and (B) real contact line length ($L_g$ and $L_r$) versus normal force for different friction coefficients and stratum corneum moduli. The gross contact line length grows in a staircase-like manner with increasing contact force. Sudden jumps in the graphs indicate that two new ridges have come into contact with the surface (one on each side, due to the perfect symmetry of our fingertip model). The inclusion of new ridges causes smoother increments in the real contact line length since only the actual contacted lengths are counted, and not the non-contacting gaps between fingerprint ridges. At the sharp increment instances, the contact line lengths computed for the cases involving friction momentarily exceed the ones obtained for the same $E_{sc}$ without friction. The smooth black curves in the plots are obtained by fitting Eq (1) to the simulated contact line results multiplied by 10 mm to represent the perpendicular length of the finger pad that is assumed in the simulations. We plot the fits only for $\mu$ = 2.0 mm$^{-1}$ since these curves overlap closely with the ones obtained for $\mu$ = 0.0 mm$^{-1}$. The fitting parameters computed for the simulated gross and real contact areas are given in Table 1 ($R^2 > 0.98$). Similar to the experimental results, the fits for the gross contact areas have comparatively higher $k$ and lower $m$ values. In addition, changing the stratum corneum modulus strongly influences $k_r$, resembling the relatively larger variation of this parameter across different hydration conditions (Fig 2B).

Next, we analyze the contact line lengths for the maximum pressing force of 1 N. Fig 4 shows the (C) gross and (D) real contact line lengths as a function of stratum corneum modulus and friction coefficient; the surfaces highlight the trends in the final contact line lengths in Fig 4A and 4B, respectively. A 20-fold decrease in the stratum corneum modulus only slightly increases the terminal breadth of the gross contact line while approximately doubling the final real contact line length. The influence of friction (relative to $E_{sc}$) on the terminal values is greater for $L_g$ compared to $L_r$. To facilitate their interpretation, the results shown in Fig 4C and 4D are also plotted as 2D curves in S2A and S2B Fig, respectively.

## Discussion

This study investigates the initial evolution of contact between the human fingertip and a flat glass surface through experimental and computational methods. During the experiments, the physical conditions were altered by hydrating the finger skin with water or glycerin. We performed high-fidelity finite element simulations with varied parameters to interpret the

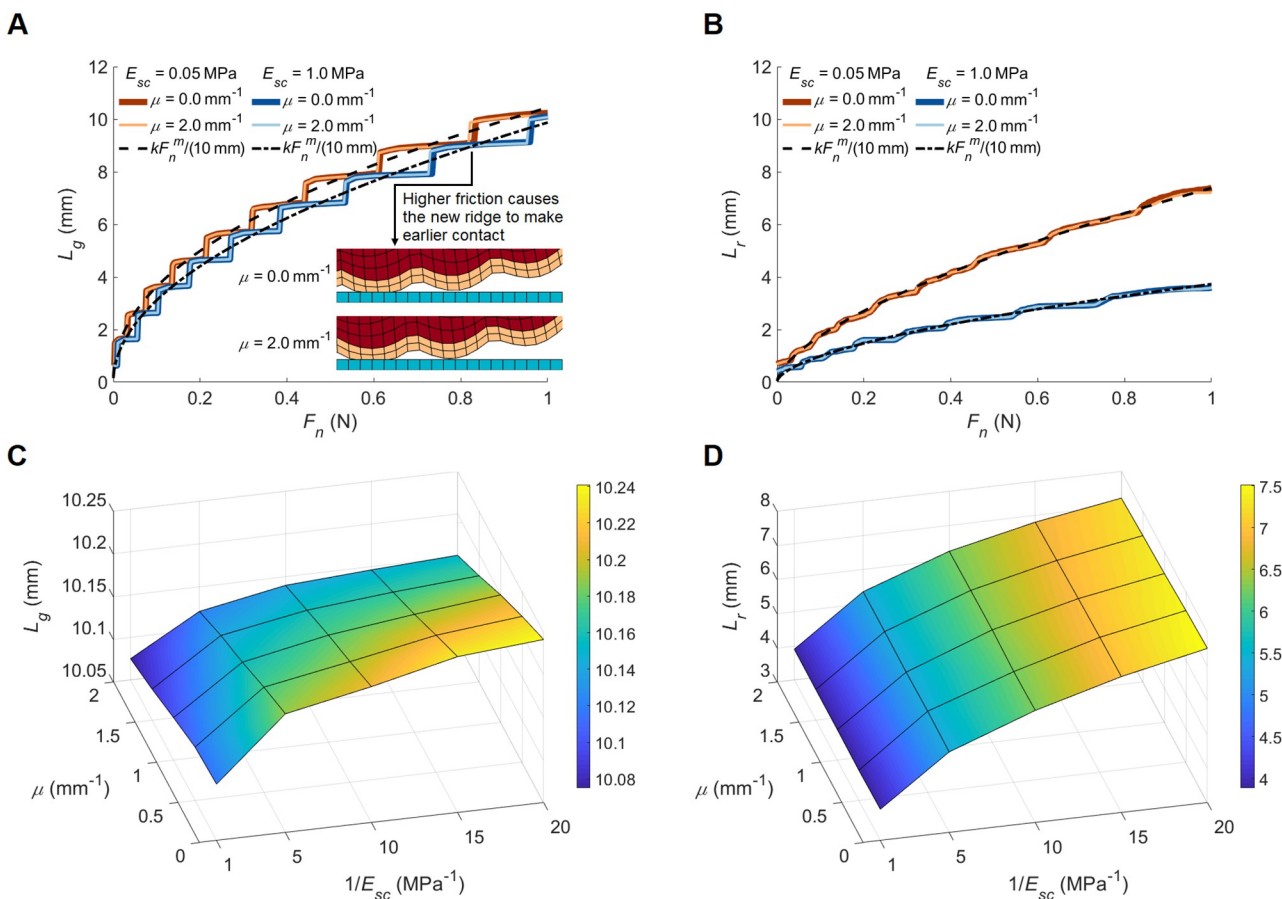

**Fig 4. Contact line length results from the finite element analyses.** (A) gross and (B) real contact line length vs. pressing force for different modified friction coefficient ($\mu$) and elastic modulus ($E_{sc}$) values. Final (C) gross and (D) real contact line lengths as a function of friction and stratum corneum modulus for 1.0 N pressing force.

**Table 1. Coefficients and exponents for the $A = kF_n^m$ curves fitted to the simulation results.** The computed contact lines are multiplied by 10 mm (the assumed elemental thickness in the simulations) to calculate the corresponding contact areas used in the equation. The units of $k$ are $mm^2N^{-m}$, while $m$ is unitless.

| | $E_{sc}$ = 0.05 MPa | | $E_{sc}$ = 1.0 MPa | |
|---|---|---|---|---|
| | $\mu$ = 0.0 mm$^{-1}$ | $\mu$ = 2.0 mm$^{-1}$ | $\mu$ = 0.0 mm$^{-1}$ | $\mu$ = 2.0 mm$^{-1}$ |
| $k_g$ | 104.5 | 104.4 | 98.6 | 98.8 |
| $m_g$ | 0.46 | 0.46 | 0.50 | 0.50 |
| $k_r$ | 74.7 | 73.8 | 37.4 | 37.3 |
| $m_r$ | 0.64 | 0.62 | 0.58 | 0.58 |

experimental results and further investigate underlying mechanisms during contact. The results give valuable insights into the influence of hydration, tissue elasticity, and friction on skin mechanics.

Hydration by glycerin resulted in the rapid development of real contact area, $A_r$, to strikingly higher values than the other conditions for both subjects. However, water did not cause the same effect; it only slightly increased $A_r$ of S1. Interestingly, neither water hydration nor

glycerin hydration caused a drastic difference in the evolution of the gross contact area. The plasticizing effect of water and glycerin on the stratum corneum is well known [30, 31]. A lower elastic modulus means softer skin, causing a larger area during contact on the glass (compare the results for different elastic modulus values in Fig 4). Hence, diminished elastic modulus can explain the differences in the $A_r$ values for hydration with glycerin. However, the remarkable discrepancy in the $A_r$ values between the water and glycerin conditions may not only mean that glycerin softens the stratum corneum more than water. It is possible that the viscosity and adhesive properties of glycerin caused more liquid bridges [21] to occur compared to water (see glycerin and water in Fig 2). The absence of the liquid bridges could be explained by the evaporation of water via the microfluidic capillary evaporation mechanism of the fingerprint ridges [9, 18] (see condensation in raw images in Fig 2).

The simulated gross and real contact line curves qualitatively agree with the experimental results (compare Fig 4A and 4B with Fig 2 and S1 Fig). Similar to the area measurements for different hydration conditions, the real contact line experiences significantly greater length changes in comparison with the gross contact line when the stratum corneum modulus is reduced. The substantial influence of the stratum corneum softening on the real contact line length originates from the flattening of the ridges, which reduces the gaps between the contact regions. The softening affects the gross contact line length differently, as it causes the leaps to occur at lower normal forces but induces only a small change in the overall magnitudes.

Friction also plays a role in normal touch as it resists the horizontal expansion of the contact line caused by the Poisson effect. However, the simulation results show that the general effect of friction on the contact development line is small compared to the influence of the elastic moduli. This difference is reasonable since the total tangential force is low, as there is no bulk lateral finger movement. However, when the stratum corneum is softer, friction exhibits a greater influence on the contact line curves for larger normal force values. Increasing pressing force induces larger tangential skin displacement along the surface, thus producing greater friction forces. For $E_{sc} = 0.05$ MPa and $\mu = 1.0$ mm$^{-1}$, the sum of the absolute values of the nodal friction forces is 0.071 N, which is around 1.5 times the total frictional force applied to the typical finger at the same friction level, as seen in Fig 3B. At the fully pressed state, friction has a stronger influence (relative to stratum corneum modulus) on the gross contact span compared to the real contact line length.

Moreover, increasing friction causes the jumps in gross contact line length to occur slightly earlier (at lower normal force) regardless of elastic modulus (Fig 4A). Hence, the high-friction contact line lengths overtake the low-friction ones for a narrow force interval, even though the latter are slightly larger in the relatively flat regions where no additional ridges come into contact. This phenomenon originates from the curved and wavy shape of the elastic body, and we name it friction-induced hinging since increasing friction counteracts the slip of the last contacting peak so that the new peak can more easily hinge over it and make contact. Such an effect is counter-intuitive since friction is normally thought to decrease the contact area during normal touch [25, 32]. Effects of friction-induced hinging can also be observed in the real contact line length curves (Fig 4B).

We also check our initial assumption on the finger depth perpendicular to the cross-section. For the nominal stratum corneum modulus ($E_{sc} = 1.0$ MPa) and medium friction ($\mu = 1.0$ mm$^{-1}$), $L_g$ at the maximum pressing force (1 N) is calculated as 10.1 mm. This result indicates that our assumption of 10 mm depth for the finite element model is fairly accurate, since the contact area at 1 N was previously found to be around 1 cm$^2$ in several available studies [33]. Nevertheless, our FE model has some limitations. The cross-section model was developed exclusively for analyzing a fingertip that is horizontally pressed into a flat surface with a contact angle of approximately 0˚. In addition, for fast contact scenarios, the model's agreement with

experimental results is likely to deteriorate since the simulation does not include inertial or viscous forces. Higher pressing forces can also reduce the accuracy of FEA since the elastic properties of the tissue can change due to stress-induced effects.

Although the experimental conditions were almost identical, the gross and real contact areas of S1 and S2 differ from one another (see Fig 2A and 2B). This variability can be caused by several factors, such as gender, age, finger size, finger pad topography, and skin mechanical and chemical properties [5, 34]. Despite this variability, for both S1 and S2, the proportion of real to gross contact area shows a distinct effect of glycerin hydration on fingertip contact compared to other conditions (see Fig 2A). Future experiments should investigate how well this rule holds for other individuals.

Our findings indicate that the type of liquid used for hydration significantly influences the resultant finger contact area evolution. This information signals another parameter to consider for understanding human grip and suggests that humans may vary their force control strategies to ensure stable grip when their fingers have been exposed to different compounds (e.g., water, sweat, soap, hand cream, sanitizer, oil). It also gives valuable insights for the design and control of future soft artificial fingers. For instance, producing robotic or prosthetic fingers from a material that changes elastic modulus as a systematic function of hydration can ensure a stable grasp of wet objects. While using such a technology, different hydration conditions, such as moist or oily contacts, possibly need a customized control strategy. We also remark that the sudden jumps in the contact area development and the discovered friction-induced hinging phenomenon can play a role in adjusting the grasping force of grippers with soft grooved contact surfaces. In addition to robotics, our findings may potentially interest researchers and product developers working in surface haptics or the cosmetics industry. For example, when people interact with a tactile display using their dry or moist fingers, they may feel the same haptic cues differently due to dissimilar fingertip deformations [35]. Hence controlling these displays based on contact area measurement may enable more consistent interaction. Finally, real contact area evolution seems to be a good metric for testing the softening effect of cosmetic creams.

## Materials and methods

### Experimental data acquisition

The finger pad contact images were measured by an apparatus similar to the ones used in earlier studies [19, 28, 36]. A glass surface (SCT3250, 3M Inc.) was mounted on top of two force sensors (Nano 17 Titanium SI-16–0.1, ATI Inc.). The two contact force vectors were measured by a data acquisition board (PCIe 6321, NI Inc.) at a sampling rate of 10 kHz and summed together to yield the total contact force vector. A high-speed and high-resolution camera (MotionBLITZ EoSense mini2, Mikroton) and a lens (LM50HC, Kova) were installed below the glass surface to measure the finger pad contact area.

The light intensity contrast between the contact area and non-contact finger pad areas was emphasized by using the prism-based frustrated total internal reflection (FTIR) principle [19, 28, 37]. A prism (N-BK7 Right-Angle Prism Uncoated, Thorlabs Inc.) was attached beneath the glass surface with cured polydimethylsiloxane (PDMS) [36]. The contact surface was illuminated from below by a light source (OSL2, Thorlabs Inc.). To obtain homogenous illumination, the light was diffused by a collimation package (OSL2COL, Thorlabs Inc.) and a diffuser (ED1-S20-MD, Thorlabs Inc.). Also, a polycarbonate diffuser sheet (L80P1–12, Acal BFi Inc.) was attached on the prism side exposed to the direct light. The final resolution of the images was 56.3×56.3 μm/pixel. The fingerprint images were recorded at a frame rate of 261 fps by triggering the camera through a data acquisition board (PCIe 6323, NI Inc.). Data collection

from the force sensors and the camera was simultaneously triggered via MATLAB. The force data were collected via MATLAB, and the images were gathered via Motion BLITZ software. The real-time force measured in the normal direction was displayed on an LCD screen behind the experimental setup.

The experimental procedures were conducted in accordance with the Declaration of Helsinki and approved by the Ethics Council of the Max Planck Society under protocol number 18–02A. One man (age 31) and one woman (age 32) participated in the experiments; both gave written informed consent. Before the experiment, the participants washed their hands with soap and water and then dried them at room temperature. Their index fingers and the glass surface were cleaned with alcohol and allowed to dry before each measurement. The participants conducted experiments in three different finger conditions: dry, hydrated with water, and hydrated with glycerin. During the dry condition, no treatment was applied to the finger. During the hydrated conditions, a 0.002 ml water or glycerin droplet was applied to the index finger of the subject. Then, they moved their finger on a separate piece of glass until the excess liquid was absorbed, which took about 10 seconds.

During the experiment, the participant sat in front of the experimental setup. They were instructed to wait until they heard an audio cue and then press their index finger at the center of the glass surface horizontally with a zero-degree angle. With the help of a visual indicator shown on the screen, the participant gradually increased their pressing force up to 1 N over a time window of 5.76 seconds. The finger pad images and contact force vector were captured automatically starting 0.1 seconds after triggering the audio cue. The recording of the images was stopped 5.76 seconds after the trigger signal, regardless of the final normal force value.

Each participant completed a training session before experimentation. The training session enabled subjects to adjust the finger angle and normal force to the desired values. They completed the experiment in three sessions separated by condition. The duration of each session was about 10 minutes.

### Image and force vector processing

Since the participants actively interacted with the glass surface during the experiments, the recorded normal force data was slightly affected by finger trembling, causing subtle vibrations in the measurements (see raw pressing force traces in Fig 2A). A preliminary analysis showed that the contact area is not significantly affected by these subtle vibrations. Hence, for clarity of the figures, the contact force vector collected in the normal direction was first low-pass filtered by a third-order zero-phase Butterworth filter with a cut-off frequency of 1 Hz. Then, it was resampled at the frame rate of the camera using interpolation so that we could calculate the evolution of gross and real contact area as a function of applied force.

The images collected by the camera do not directly represent the correct finger pad contact, as they are distorted by the camera's intrinsic, extrinsic, and lens distortion parameters. Hence, the images were calibrated before calculating the finger pad contact area. The intrinsic and lens distortion parameters were calculated by collecting several images of a checkerboard from different angles and then processing those images using the MATLAB Camera Calibration Toolbox [38]. Then, the extrinsic parameters due to the camera viewing angle were calculated by capturing an image of a 14×14 mm PDMS square on the glass surface and calculating the transformation matrix that converts the viewed shape to a square using the Hough transformation method [39]. We used the same image to calculate the area per pixel (56.3×56.3 μm/pixel) by dividing the actual area of the square by the number of pixels it occupied in the transformed image.

After the calibration process, the finger pad images were processed by following the procedures explained by Lévesque and Hayward [28]. First, the images were smoothed by a Gaussian filter to reduce noise. Then, the local average in a square window with a width of 20 pixels was computed for each pixel in each image. High local average and low local variance indicate the background pixels, which do not contain finger contact [40]. A binarization operation was conducted to identify contact pixels (black) using the local average map as a pixel-wise threshold on the foreground image. This local adaptive threshold methodology prevented the misdetection of condensation (see Fig 1) as finger contact.

The real contact area of the finger pad in each image was calculated by summing the number of black pixels in the binarized image and multiplying them with the calibrated pixel area value. For the gross contact area of the whole finger pad, the area of the polygon representing the contour of the binary fingerprint was taken. For the real contact line, the pixels representing the line passing through the fingerprint centroid perpendicular to the longitudinal finger axis were summed. The gross contact line was calculated as the distance between the largest and smallest pixel index on that contact line.

## Simulations

We developed in-house FEA software with MATLAB to perform the simulations. Within this framework we use four-noded 2D iso-geometric elements. Since the finger dimension in the longitudinal direction is larger than the height and the width of its cross-section, we utilized the plane strain deformation theory. The details of the adopted FE formulation can be found in Liu and Quek [41]. We built a set of parametric tools that together are capable of generating discretized finger models for various geometries, performing calculations, extracting the desired physical quantities, and visualizing the results.

**Geometry and mesh.**  The finger model is constructed using the following four main deformable skin layers attached to the rigid phalanx bone: subcutaneous tissue (hypodermis), dermis, epidermis, and stratum corneum. Fig 5A shows a schematic of the model, which has an elliptical shape before deformation. The upper surface includes a fingernail in place of the two outermost skin layers. The stratum corneum's fingerprints are modeled as a positivized sine wave perpendicular to the lower skin surface. The model also includes the details of the wavy interface between the epidermis and dermis layers. The nominal model dimensions are

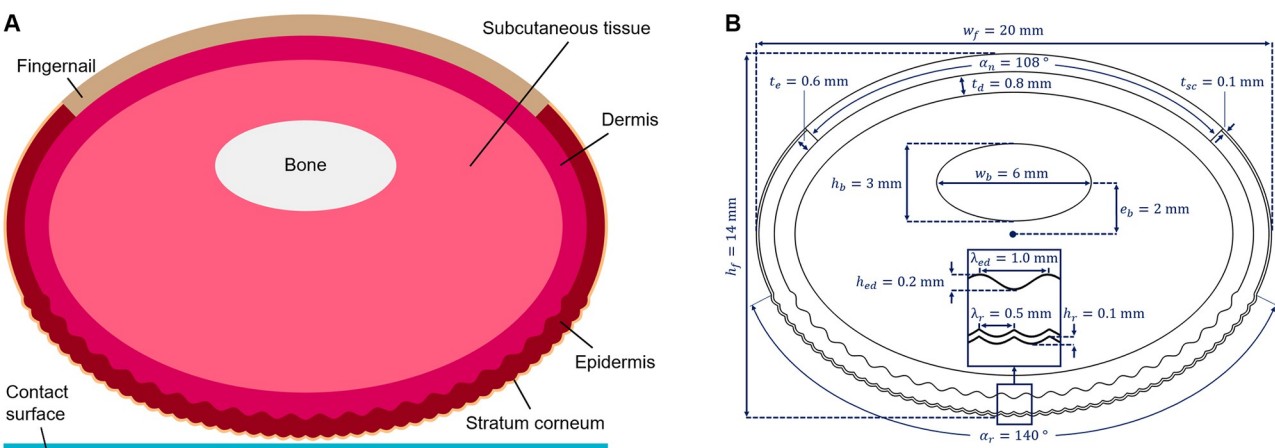

**Fig 5. Details of the fingertip model.** (A) Finite element model of the finger cross-section composed of the stratum corneum, epidermis, dermis, subcutaneous tissue, bone, and fingernail. Elemental edges in the deformable tissue are hidden for clarity. (B) The dimensions of the model.

**Table 2. Nominal values for fingertip layer elastic constants.**

|  | Stratum corneum | Epidermis | Dermis | Subcutaneous tissue | Fingernail |
|---|---|---|---|---|---|
| **Elastic modulus (MPa)** | 1.0 | 0.140 | 0.080 | 0.034 | 170.0 |
| **Poisson's ratio** | 0.3 | 0.4 | 0.4 | 0.48 | 0.3 |

taken from the most relevant studies in the literature [42, 43] and are presented in Fig 5B. We assume the fingertip depth perpendicular to the cross-section to be 10 mm, and we validate this assumption.

In our framework, we generate the elements directly to obtain a mesh with a certain pattern, rather than generating surfaces and then applying discretization. This approach ensures good element quality throughout the mesh (i.e., low element skewness) and avoids over-stiff triangular elements. The geometry and mechanical properties of each layer are separately configurable, as is the element size. We use a very fine discretization resolution in the stratum corneum, epidermis, and outer dermis to achieve detailed results near contact (see Fig 3). To maintain good element aspect ratios and reduce computational complexity far from contact, we increase the element size in the middle of the dermis layer and near the edge of the subcutaneous tissue by bridging three elements into one; this procedure results in 15,120 nodes and 14,670 elements for the finger section. The surface being contacted is modeled as a single undeformable block that is perfectly flat.

**Material properties.** We assign appropriate material properties (elastic modulus and Poisson's ratio) to each layer of the fingertip. The nominal values are taken from the literature [42] and are given in Table 2. The elastic modulus of human bone [44] is three orders of magnitude larger than that of the fingernail, which is the stiffest of the five deformable materials in the finger cross-section. Therefore, we consider the bone as rigid, and the nodes on the interface between the bone and the subcutaneous tissue are fixed to enforce this condition numerically.

**Contact and friction modeling.** The simulations need to use a very small step size to accurately capture the contact mechanics of interest. Therefore, the finger model is moved toward the surface in increments of 0.004 mm, which is around half of the edge length of a stratum corneum element. When a node penetrates into the surface, contact is detected, and a contact force vector is created. Then, we optimize all of the contact force vectors to push back all of the contacting nodes such that they are barely touching the surface. In this optimization subroutine, the convergence tolerance value has been chosen as half of the displacement step size (0.002 mm).

Friction between the skin and the surface is implemented in the simulations using a modified Coulomb model [45] that combines the principles of the Coulomb and Dahl friction models [46], acting like a lateral spring that tries to return the node to the location where it initially contacted the surface. We calculate the tangential force ($F_t$) on each node by combining the influence of the normal force component ($F_n$) and the tangential displacement ($d_t$) as:

$$F_t = -\mu d_t F_n \qquad (2)$$

where $\mu$ is the modified friction coefficient with units of inverse length. The tangential force always opposes any tangential displacement that has occurred in a spring-like manner, with a spring stiffness that is the product of $\mu$ and $F_n$. Since the tangential displacements change with the applied tangential forces, the equilibrium solution is calculated iteratively.

**Analysis procedure.** The described framework enables us to simulate the movement of the model's 15,120 nodes as it presses straight into a flat, rigid surface. We start the simulation

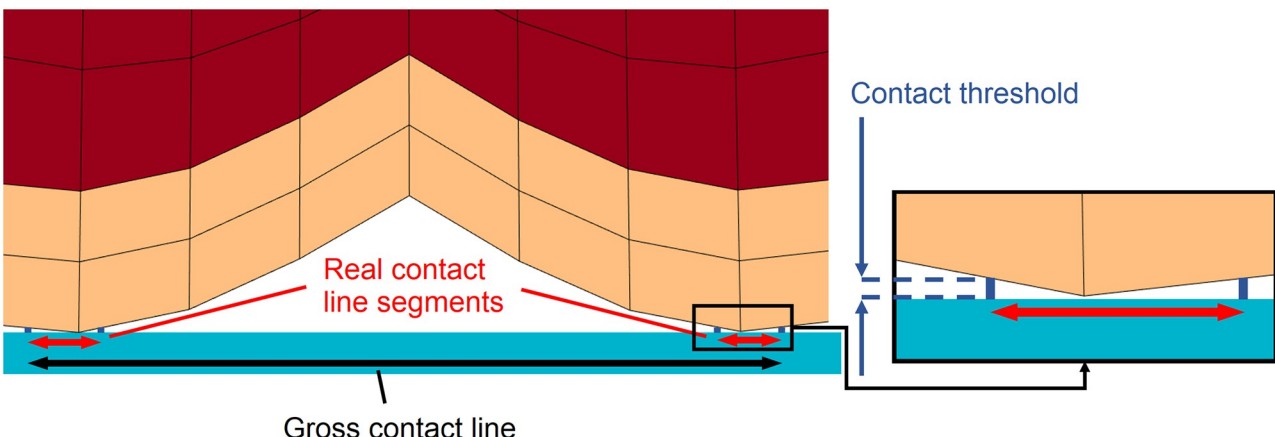

**Fig 6. FEA-based calculation of the gross and real contact lines.** The gross contact line length is determined as the length of the line that connects the leftmost and rightmost points in contact. The real contact line length is calculated by summing up the lengths of all contact line segments.

by applying vertical displacement increments to the entire finger, whose lowermost nodes are initially located just above the surface. The finger moves downward freely until the first contacting nodes are detected. Then, a quasi-static solution is calculated at each step, and all outputs are recorded, including each node's normal force, each node's frictional force, and the gross and real contact line length (discussed below). The simulation is terminated when the total normal force reaches a limit value of 1.0 N. S1 Video shows a video of the fingertip model pressing into the surface for the nominal stratum corneum modulus ($E_{sc}$ = 1.0 MPa) and medium friction ($\mu$ = 1.0 mm$^{-1}$).

**Contact line calculation.** We calculate the line of contact between the skin and the surface based on the following contact detection principle. Since a node is considered to be in contact when its distance to the surface is smaller than a tolerance value of 0.002 mm, this same value is used for detecting the portions of the external element edges that are in contact with the surface. For this task, we finely discretize the boundary of the finger such that each elemental edge is divided into approximately 50 segments, and additional points are created between the elemental nodes. Discretization is also applied to the surface with the same resolution. Then the surface points that are in contact with at least one finger point are identified.

In the simulations, the gross contact line is determined by simply calculating the length of the line segment bounded by the leftmost and rightmost finger points in contact. We calculate the real contact line by first determining the starting and ending points of all of the contact intervals. Then, the lengths of the contact intervals are summed. The contact line calculation principle is depicted in Fig 6. The gross contact line is always larger than or equal to the real contact line, and both are measured in millimeters.

## Supporting information

**S1 Fig. Results of the contact line calculations from experimental data.** The evolution of gross and real contact lines as a function of applied force for both subjects (S1 and S2 Figs). The results for the three hydration conditions are color-coded.
(TIF)

**S2 Fig. Simulated contact line length vs. inverse stratum corneum modulus ($1/E_{sc}$) curves for different friction coefficients ($\mu$) and 1.0 N pressing force.** The results obtained for (A) gross and (B) real contact lines are equivalent to the surface plots presented in Fig 4C and 4D,

respectively.
(TIF)

**S1 Video. A video of the fingertip model pressing into the surface.** Nominal stratum corneum modulus ($E_{sc}$ = 1.0 MPa) and medium friction ($\mu$ = 1.0 mm$^{-1}$) are used in this example case.
(MP4)

## Acknowledgments

The authors thank Shao Wen Wu for designing the experimental apparatus, Saekwang Nam for providing helpful comments on the setup design, Amirreza Aghakhani for helping attach the prism to the glass surface using PDMS, and Luzia Knoedler for helping with image processing. Gokhan Serhat and Yasemin Vardar contributed equally to this work.

## Author Contributions

**Conceptualization:** Gokhan Serhat, Yasemin Vardar, Katherine J. Kuchenbecker.

**Formal analysis:** Gokhan Serhat, Yasemin Vardar.

**Investigation:** Gokhan Serhat, Yasemin Vardar.

**Methodology:** Gokhan Serhat, Yasemin Vardar, Katherine J. Kuchenbecker.

**Software:** Gokhan Serhat, Yasemin Vardar.

**Supervision:** Katherine J. Kuchenbecker.

**Visualization:** Gokhan Serhat, Yasemin Vardar.

**Writing – original draft:** Gokhan Serhat, Yasemin Vardar, Katherine J. Kuchenbecker.

**Writing – review & editing:** Gokhan Serhat, Yasemin Vardar, Katherine J. Kuchenbecker.

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
