## [Decision Letter · Decision Letter 0]

24 Jan 2022

PONE-D-21-37465Contact evolution of dry and hydrated fingertips at initial touchPLOS ONE

Dear Dr. Serhat,

Thank you for submitting your manuscript to PLOS ONE. After careful consideration, we feel that it has merit but does not fully meet PLOS ONE’s publication criteria as it currently stands. Therefore, we invite you to submit a revised version of the manuscript that addresses the points raised during the review process.

We look forward to receiving your revised manuscript.

Kind regards,

Edoardo Sinibaldi

Academic Editor

PLOS ONE

Journal Requirements:

2. Please note that PLOS ONE has specific guidelines on code sharing for submissions in which author-generated code underpins the findings in the manuscript. In these cases, all author-generated code must be made available without restrictions upon publication of the work. Please review our guidelines at https://journals.plos.org/plosone/s/materials-and-software-sharing#loc-sharing-code and ensure that your code is shared in a way that follows best practice and facilitates reproducibility and reuse. Code may be shared by providing a URL within the Methods section to a code repository or it may be uploaded as a supplemental file.

Additional Editor Comments:

Both Reviewers highlighted that the study has clear points of merits and can be of interest. Moreover, both Reviewers provided constructive comments for improving the manuscript.

Reviewer1, in particular, provided a rather extended set of points that can strengthen both methodology and results (besides the whole presentation flow), thus leading to more robust claims, based on a wider statistical analysis.

The Authors are encouraged to leverage the raised comments in order to improve the scientific solidity of the manuscript.

Reviewers' comments:

Reviewer's Responses to Questions

**Comments to the Author**

1. Is the manuscript technically sound, and do the data support the conclusions?

Reviewer #1: Yes

Reviewer #2: Yes

2. Has the statistical analysis been performed appropriately and rigorously? 

Reviewer #1: No

Reviewer #2: Yes

3. Have the authors made all data underlying the findings in their manuscript fully available?

Reviewer #1: No

Reviewer #2: Yes

4. Is the manuscript presented in an intelligible fashion and written in standard English?

Reviewer #1: Yes

Reviewer #2: Yes

5. Review Comments to the Author

Reviewer #1: This study quantifies the influence of hydration on fingertip contact evolution, as probed by the gross and real contact area. Those quantities are obtained in vivo from two participants using an optical imaging system monitoring the contact over a transparent surface. It is found that hydration has little effect on the gross contact, but greatly increases the real contact area. The experimental data is then compared to a simulation mimicking the experiment. The authors varied stratum corneum stiffness and contact friction to depict which of those influence the real contact, and found that the stiffness of the SC strongly affects the real contact whereas friction does not.

This is a very interesting study, the measurements made and the comparison to the model are novel and bring new results. The methods are sound and well detailed. The conclusions are well supported by the data. Here is a few important points to consider to improve the paper:

- The introduction is hard to follow and misuses several reference references: for instance, some references used for the normal contact are actually experiments on tangential loading and vice versa (for example ref [9][14][15], at L27). See also comments below. I would recommend the authors to try to improve it.

- Fig2A: please add the corresponding normal force traces. The curves shown in fig 2A are very likely influenced by the speed at which the finger is loaded, it is, therefore, essential to show that the loading dynamics were the same across conditions. Some numbers about the rate (L121-125) are insufficient.

- The data was obtained from two subjects. Given the variability of the results across subjects, it would be valuable to double or triple the sample.

- Statistics:

- The results related to the model should be presented more coherently with respect to the experimental data. This part lacks clear quantification. Can’t you fit the data to the same eq as the experimental data (eq 1), and compare the effect of friction and softening in the model to those observed in vivo (k and m coefficients)? Moreover, the 3D plots are hard to interpret.

- Please confirm that those t-tests were corrected for multiple comparisons. Why not use an ANOVA test since there are three factors, or even better, using a mixed-effect model and including both subjects in the same analysis?

- The discussion should better discuss the limitation of the approach, in particular, the limitation related to the model, which assumes many simplifying hypotheses.

- Finally, I would tone down the comments made with respect to the “friction-induced hinging” phenomenon. First, it results from a particular configuration of the model (gradual contact of the fingerprints) that cannot be reproduced in vivo (the fingerprint are continuously increasing the contact). Second, the effect size of the phenomenon seems very small and is not even quantified.

Other minor points:

L3-4: not sure those are the right references to support the statement

L11-13: frictional force? Or friction coefficient?

L16: ref 8 is about tactile sensors, not about skin.

L21: maybe add Fingerprint ridges allow primates to regulate grip, Yum, et al. PNAS 117 (50), 31665-31673

L24: [15] is also during the onset of slip

L27: How about Initial contact shapes the perception of friction, Willemet et al. PNAS 118 (49)

L71: “The simulation results showed that the elastic moduli influence both gross and real contact area more significantly than friction”. It is unclear here why would friction influence real contact.

L75: “Around these force values, we also detected an interesting physical phenomenon that we name friction-induced hinging, which causes the gross contact line length to be momentarily greater than the one observed for lower friction”, This is very unclear at this point, even if it is explained later on.

L84: “at a fixed contact angle”: please give a value

L105: “The resulting fits (R 2 > 0.93) appear with the data in Fig 2A”. I am not sure that is true.

L115: “Independent t-tests showed that the k r values for the glycerin condition were significantly higher than the other conditions for both subjects ( p < 0 . 05)”. See major comment.

L181-187: This part lack of clear quantification. Can’t you fit the data to the same eq as the experimental data (eq 1), and compare the effect of friction and softening in the model to those observed in vivo (k and m coefficients)? Moreover, the 3D plots are hard to interpret.

L208-210: “The absence of the liquid bridges could be explained by the evaporation of water via the microfluidic capillary evaporation mechanism of the fingerprint ridges [7,19] (see condensation in raw images in Fig 2).” This is worrying, how did you compensate for it?

L321: “The contact force vector collected in the normal direction was first low-pass filtered by a third-order zero-phase Butterworth filter with a cut-off frequency of 0.5Hz. Then, it was resampled at the frame rate of the camera using interpolation.” Cutoff frequency seems extremely low, please justify.

Reviewer #2: This manuscript presents very nice work on the evolution of contact as a finger pad presses against a flat, rigid surface. Results are presented for both real measurements and simulations, under different conditions of hydration. Real measurements are taken using a sophisticated apparatus and image processing that detects contact by frustrated total internal reflection. Skin hydration is varied by either applying no additional hydration or applying a drop of water or glycerin. A similarly sophisticated finite element simulation is used to measure contacts along a line for a wider range of friction conditions. The results of both approaches are in agreement, and complement each other. Interesting findings are reported, including a new phenomenon that the authors call friction-induced hinging.

I’m impressed by the quality of the work as well as its clear and detailed presentation. The manuscript presents a substantial amount of work, including the development of sophisticated methodologies for both real measurement and simulations. The work is rigorous and well analysed. The prior work is clearly described. The discussion is reasonable, and clarifies the implications of the findings and the connections between the real and simulated results. I think this is an excellent paper and I have very few improvements to suggest:

1. It would be interesting for the reader to better understand the significance of water and glycerin as hydration mechanism for the fingerpad. I assume, for example, that the hydration of the fingerpad often varies due to the presence or build-up of sweat. Would sweat be considered closer to water or glycerin? Are there other situations of practical interest, such as having oily fingers as noted in the discussion?

2. Line 242: “can also observed” is missing a “be”.

3. Figure 2A: Should the labels be S1, S2 instead of SA, SB?

6. PLOS authors have the option to publish the peer review history of their article (what does this mean?). If published, this will include your full peer review and any attached files.

Reviewer #1: No

Reviewer #2: No

---

## [Author Response · Author response to Decision Letter 0]

26 Mar 2022

Dear Reviewers,

We sincerely thank you for your thoughtful and constructive comments on our submitted manuscript. We carefully considered all of your feedback and revised our paper accordingly. You can find our responses to the specific comments in the response to reviewers document.

We believe these changes have made our manuscript significantly stronger, and we thank you again for your helpful input.

Kind regards,

The Authors

---

## [Decision Letter · Decision Letter 1]

27 May 2022

Contact evolution of dry and hydrated fingertips at initial touch

PONE-D-21-37465R1

Dear Dr. Serhat,

We’re pleased to inform you that your manuscript has been judged scientifically suitable for publication and will be formally accepted for publication once it meets all outstanding technical requirements.

Kind regards,

Edoardo Sinibaldi

Academic Editor

PLOS ONE

Additional Editor Comments (optional):

All the raised comments have been positively addressed, thus strengthening the scientific solidity of the revised manuscript.

Reviewers' comments:

Reviewer's Responses to Questions

**Comments to the Author**

1. If the authors have adequately addressed your comments raised in a previous round of review and you feel that this manuscript is now acceptable for publication, you may indicate that here to bypass the “Comments to the Author” section, enter your conflict of interest statement in the “Confidential to Editor” section, and submit your "Accept" recommendation.

Reviewer #1: All comments have been addressed

Reviewer #2: All comments have been addressed

2. Is the manuscript technically sound, and do the data support the conclusions?

Reviewer #1: Yes

Reviewer #2: Yes

3. Has the statistical analysis been performed appropriately and rigorously? 

Reviewer #1: Yes

Reviewer #2: Yes

4. Have the authors made all data underlying the findings in their manuscript fully available?

Reviewer #1: No

Reviewer #2: Yes

5. Is the manuscript presented in an intelligible fashion and written in standard English?

Reviewer #1: Yes

Reviewer #2: Yes

6. Review Comments to the Author

Reviewer #1: All my comments have been thoroughly and adequately addressed.

I have not further concerns. Thank you!

Reviewer #2: My original review as Referee 2 was already very positive, with a few minor requests for corrections and clarifications. My comments were addressed satisfactorily in the revised manuscript.

I also reviewed the more substantial recommendations made by Referee 1, as well as the changes made by the authors to address them. I believe that the manuscript has been greatly improved by taking these comments into account. Two points of contention may remain:

1. Number of participants: While an increase in the number of participants (currently 2) would be an improvement, it doesn’t seem abnormally low to me for a study of this type. I do believe that the current results are sufficient for publication and I understand that repeating this experiment is no longer feasible for the authors.

2. Hinging phenomenon: I agree with the authors that the hinging phenomenon is worth mentioning and that there is enough support for this claim in the paper. I believe that toning down the text related to this finding was a reasonable compromise.

7. PLOS authors have the option to publish the peer review history of their article (what does this mean?). If published, this will include your full peer review and any attached files.

Reviewer #1: No

Reviewer #2: No

---

## [Editor Report · Acceptance letter]

16 Jun 2022

PONE-D-21-37465R1 

Contact evolution of dry and hydrated fingertips at initial touch 

Dear Dr. Serhat:

I'm pleased to inform you that your manuscript has been deemed suitable for publication in PLOS ONE. Congratulations! Your manuscript is now with our production department. 

Kind regards, 

on behalf of

Dr. Edoardo Sinibaldi 

Academic Editor

PLOS ONE